# Embryonic Thermal Manipulation Affects the Antioxidant Response to Post-Hatch Thermal Exposure in Broiler Chickens

**DOI:** 10.3390/ani10010126

**Published:** 2020-01-13

**Authors:** Khaled M. M. Saleh, Amneh H. Tarkhan, Mohammad Borhan Al-Zghoul

**Affiliations:** 1Department of Applied Biological Sciences, Faculty of Science and Arts, Jordan University of Science and Technology, P.O. Box 3030, Irbid 22110, Jordan; khaledmousa93@gmail.com (K.M.M.S.); amneht92@gmail.com (A.H.T.); 2Department of Basic Medical Veterinary Sciences, Faculty of Veterinary Medicine, Jordan University of Science and Technology, P.O. Box 3030, Irbid 22110, Jordan

**Keywords:** broiler, thermal manipulation, antioxidant, heat stress, cold stress

## Abstract

**Simple Summary:**

The broiler chicken is one of the most important livestock species in the world, as it occupies a major role in the modern human diet. Due to uneven artificial selection pressures, the broiler has increased in size over the past few decades at the expense of its ability to withstand oxidative damage, the latter of which is often a byproduct of thermal stress. In order to attenuate the effects of heat stress, thermal manipulation (TM), which involves changes in incubation temperature at certain points of embryonic development, is increasingly being presented as a way in which to improve broiler thermotolerance. Therefore, the objective of this study was to investigate how TM might affect broiler response to post-hatch thermal stress in the context of the genes that help combat oxidative damage, namely the catalase, NADPH oxidase 4 (*NOX4*), and superoxide dismutase 2 (*SOD2*) genes. Expression of all three aforementioned genes differed significantly between TM and control chickens after exposure to cold and heat stress. Conclusively, TM may act as a viable mode of preventative treatment for broilers at risk of thermally induced oxidative stress.

**Abstract:**

Thermal stress is a major source of oxidative damage in the broiler chicken (*Gallus gallus domesticus*) due to the latter’s impaired metabolic function. While heat stress has been extensively studied in broilers, the effects of cold stress on broiler physiologic and oxidative function are still relatively unknown. The present study aimed to understand how thermal manipulation (TM) might affect a broiler’s oxidative response to post-hatch thermal stress in terms of the mRNA expression of the catalase, NADPH oxidase 4 (*NOX4*), and superoxide dismutase 2 (*SOD2*) genes. During embryonic days 10 to 18, TM was carried out by raising the temperature to 39 °C at 65% relative humidity for 18 h/day. To induce heat stress, room temperature was raised from 21 to 35 °C during post-hatch days (PD) 28 to 35, while cold stress was induced during PD 32 to 37 by lowering the room temperature from 21 to 16 °C. At the end of the thermal stress periods, a number of chickens were euthanized to extract hepatic and splenic tissue from the heat-stressed group and cardiac, hepatic, muscular, and splenic tissue from the cold-stressed group. Catalase, *NOX4*, and *SOD2* expression in the heart, liver, and spleen were decreased in TM chickens compared to controls after both cold and heat stress. In contrast, the expression levels of these genes in the breast muscles of the TM group were increased or not affected. Moreover, TM chicks possessed an increased body weight (BW) and decreased cloacal temperature (T^C^) compared to controls on PD 37. In addition, TM led to increased BW and lower T^C^ after both cold and heat stress. Conclusively, our findings suggest that TM has a significant effect on the oxidative function of thermally stressed broilers.

## 1. Introduction

The term broiler refers to any member of the red junglefowl subspecies, *Gallus gallus domesticus*, that has been reared for the purposes of meat production and consumption [1]. Constituting the largest standing avian population, the broiler has become a vital component of modern human nutrition as a result of the rapid industrialization of the poultry production process [2]. Since the mid-twentieth century, broilers have undergone intensive breeding in order to enhance their growth rates, meat yield, and feed conversion ratios [3]. However, these artificial selection pressures have been largely consumer-driven, focusing solely on improving certain commercially attractive parameters at the expense of immune, metabolic, and skeletal function [4]. As a result, the modern broiler has become increasingly susceptible to the effects of thermal and, in turn, oxidative stresses [5].

Due to impaired metabolic function and expensive energetics, broilers are especially vulnerable to heat stress, which occurs when the broiler is unable to adequately dissipate body heat to the environment [6]. Rising global temperatures have consolidated the threats of heat stress to the development and wellbeing of broiler chickens, and such increases in temperature are exacerbated during the hotter seasons [7]. In addition, broiler heat stress can be caused by certain stages of the poultry production process, especially during their transport from rearing to processing facilities [8]. Broilers subject to heat stress will have a lower body weight due to decreased feed intake, and their innate immune function will be impaired as a result of decreased immune organ weight [9]. In fact, it has been illustrated that heat stress results in oxidative stress in broilers, resulting in a number of adverse metabolic changes [10].

Heat stress is a major cause of oxidative stress in broilers, and oxidative damage deteriorates the appearance, flavor, and nutritional value of broiler meat [11]. Oxidative stress can be defined as the imbalance that occurs when the amount of reactive oxygen species (ROS) in an animal cell exceeds the latter’s antioxidant capacity [12]. To prevent oxidative stress, several genes are involved in the maintenance of cellular homeostasis, including NADPH oxidase 4 (*NOX4*), superoxide dismutase (*SOD2*), and catalase [13,14]. Primarily expressed in renal and vascular cells, the *NOX4* gene is constitutively active and codes for an oxygen-sensing enzyme that can also play a role in antimicrobial defense [15,16,17]. If overexpressed, *NOX4* leads to oxidative stress due to its production of superoxide (O_2_^−^) radicals and hydrogen peroxide (H_2_O_2_) molecules [18,19]. To prevent NOX4-associated oxidative stress from occurring, SOD2 and catalase act to dismutate O_2_^−^ and break down H_2_O_2_, respectively [20].

Unlike heat stress, cold stress in broilers has not been the subject of much research in the context of its relation to oxidative stress. Nonetheless, cold stress has been found to induce oxidative stress and modulate immune function in broilers while also increasing their susceptibility to necrotic enteritis and ascites development [21,22,23,24,25]. Moreover, cold stress was found to affect the thigh muscle of broilers more severely than the breast muscle, and it significantly reduced the feed intake and body weights of broilers but increased their feed conversion ratios [26,27]. As outdoor rearing systems gain more popularity, preventing cold stress will become increasingly costly to the poultry industry, and such costs often fluctuate depending on fuel prices, season, and existing heating systems [28].

To mitigate the damage caused by heat and cold stress, thermal manipulation (TM), which involves embryonic exposure to high or low temperatures, has been found to improve thermotolerance and enhance physiological parameters of broilers [13,29,30,31,32,33]. However, further research needs to be carried out in order to understand the effects of heat- and cold-induced oxidative stress in thermally manipulated broilers. Therefore, the main purpose of the present study was to investigate the effects of both cold and heat stress on the antioxidant defense mechanisms of thermally manipulated broiler chickens.

## 2. Materials and Methods

Ethical approval for all experimental procedures was obtained from the Animal Care and Use Committee at Jordan University of Science and Technology (approval # 16/3/3/418).

### 2.1. Egg Procurement and Incubation

Fertile Cobb eggs (n = 600) were obtained from local distributors based in Madaba, Jordan. Before incubation, eggs were thoroughly examined, and eggs were excluded if they displayed abnormality or damage (n = 69). The remaining eggs (n = 531) were then randomly divided into two groups, control (n = 266) and thermal manipulation (TM) (n = 265), and incubated in semi-commercial incubators (Masalles S.L., Barcelona, Spain). In the control group, eggs were incubated under standard conditions (37.8 °C and 56% relative humidity (RH)) throughout embryogenesis. In contrast, TM eggs were only incubated under standard conditions from embryonic days (ED) 1 to 9 and 19 to 21, as TM was applied from ED 10 to 18 by incubating the eggs at 39 °C and 65% RH for 18 h/day. On ED 7, candling was performed on each egg in order to exclude infertile and/or nonviable eggs.

### 2.2. Hatchery Management

On hatch day, the hatchability, which is the percentage of fertile eggs that hatch, was calculated according to the following equation: hatchability = (number of hatched chicks/total number of incubated eggs) × 100. Chicks were left in the incubator to dry for the first 24 h of their post-hatch life, after which they were transported to a special area designated for the field experiments. On post-hatch days (PD) 1 and 37, the cloacal temperatures (T^C^) and body weights (BW) were recorded, and the number of chicks that died within the whole field experimental period was noted. Dead chickens were histopathologically examined, but no significant obvious findings were reported. Before exposure to thermal stress, chicks were randomly distributed into their coops in groups of ten. In the first week, the temperature of the enclosures was kept at 33 ± 1 °C and was steadily reduced to 24 °C by the end of the third week. The RH during the rearing period was maintained within a range of 45%–52%. Water and appropriate feed were supplied to the chicks ad libitum during the whole field experiment period. On PD 8 and 20, chicks were vaccinated against Newcastle disease, and, on PD 15, the chicks were vaccinated against infectious bursal disease. The overall experimental design is illustrated in Figure 1.

### 2.3. Experiment 1: Post-Hatch Heat Exposure

On PD 26, male chicks (n = 60) were randomly selected from each of the control and TM groups to be transported to the experimental room for the induction of heat stress. On PD 28, heat stress was induced by raising the temperature of the experimental room to 35 °C and 45%–52% RH until PD 35. During this period, male chicks (n = 60) from each of the control and TM groups were subject to normal conditions (not shown in Figure 1). The experimental and rearing rooms were located on the same floor in order to mitigate transport stress. After 0, 1, 3, 5, and 7 days of heat exposure, chicks (n = 8) were randomly chosen from the control and TM groups. The T^C^ and BW of the chicks were recorded, after which they were euthanized in order to collect hepatic and splenic organs. Samples were snap-frozen on-site using liquid nitrogen, transferred to the laboratory, and stored at −80 °C.

### 2.4. Experiment 2: Post-Hatch Cold Exposure

On PD 32, chicks (n = 40) from each of the two incubation groups (control group and TM group) were randomly chosen and subdivided into four subgroups: control exposed to cold stress (CS), TM exposed to cold stress (TS), control exposed to normal conditions (CN) and TM exposed to normal conditions (TN). Cold stress was achieved by lowering the room temperature to 16 °C and 45%–52% RH from PD 32 to 37. At PD 37, BW and T^C^ were recorded for chicks (n = 5) from each subgroup, and a number of chicks (n = 5) were humanely euthanized in order to collect the liver, spleen, heart, and breast muscle organs. Samples were snap-frozen on-site using liquid nitrogen, transferred to the laboratory, and stored at −80 °C.

### 2.5. cDNA Synthesis

The Direct-Zol™ RNA MiniPrep (Zymo Research, Irvine, CA, USA) was utilized alongside TRI Reagent^®^ (Zymo Research, Irvine, CA, USA) in order to isolate total RNA from all the collected samples. The Biotek PowerWave XS2 Spectrophotometer (BioTek Instruments, Inc., Winooski, VT, USA) was employed to determine the quantity and quality of the samples, after which 2 μg of total RNA from each sample were inputted into the Superscript III cDNA Synthesis Kit (Invitrogen, Carlsbad, CA, USA) to synthesize cDNA.

### 2.6. Primer Design and Relative mRNA Quantitation Analysis by Real-Time RT-PCR

The primer sequences that were used for real-time RT-PCR analysis are listed in Table 1. Primers were taken from previous reports [34] and were designed using the PrimerQuest tool on the Integrated DNA Technologies website (Coralville, IA, USA) (https://eu.idtdna.com/pages) and the Nucleotide database on the NCBI (Bethesda, MA, USA) website (https://www.ncbi.nlm.nih.gov/nucleotide/). The QuantiFast SYBR^®^ Green PCR Kit (Qiagen, Hilden, Germany) was utilized on a Rotor-Gene Q MDx 5 plex instrument (Qiagen, USA) according to the manufacturer’s protocol. For the internal control, fold changes in gene expression were normalized against the 28S ribosomal RNA. Single target amplification specificity was ensured by the melting curve, and relative quantitation was calculated automatically by the software on the Rotor-Gene Q MDx 5 plex instrument.

### 2.7. Statistical Analysis

IBM SPSS Statistics v23.0 (IBM, USA) was used for all statistical analyses performed in the current study. The chi-squared test was used to analyze hatchability and mortality rates. T^C^, BW, and the fold changes in mRNA levels of the catalase, *NOX4*, and *SOD2* genes are portrayed as means ± SD. An independent t-test compared between the control and TM groups with respect to several parameters at each time interval (PD 1, 3, 5, 7, 9, 11, 13, 15, 19, 22, 25, 28, 30, 33 and 35). Within the treatment group itself, two-way ANOVA was also used to compare between different parameters at specific time intervals after thermal stress. Statistical significance for parametric differences was set at 0.05.

## 3. Results

### 3.1. Effect of Thermal Manipulation (TM) on Hatchability and Physiological Parameters of Broiler Chicks

No significant effect was observed in either the mortality (control = 1.7; TM = 1.7) or the hatchability (control = 85.71; TM = 83.02) rates between the control and TM groups. However, TM led to significantly lower cloacal temperatures (T^C^) on PD 1 and 37 and to higher body weights (BW) on PD 37. However, no significant change was observed in hatchling BW (Table 2).

### 3.2. Effect of Post-Hatch Heat Stress on Physiological Parameters of Thermally Manipulated Broilers

Table 3 illustrates the effects of heat stress for 7 days (PD 28 to 35) on T^C^, BW, and BW gain in the controls and TM broiler chickens. TM significantly decreased the mortality rate during post-hatch heat exposure (control = 12%; TM = 8%). Moreover, heat stress significantly increased the T^C^ in both groups, but the T^C^ of controls was significantly higher compared to TM chicks. On day 0 (PD 28) of heat stress, the BW of the TM group was significantly higher than that in controls. Similarly, the BW of controls was significantly lower compared to TM chicks on day 7 (PD 35) of heat stress, but the subgroups exposed to heat stress possessed significantly lower BW and BW gain compared to those exposed to normal conditions.

### 3.3. Effect of Post-Hatch Heat Stress on Antioxidant Enzyme mRNA Levels in Thermally Manipulated Broilers

Figure 2 represents the effects of heat stress on the hepatic and splenic mRNA levels of certain antioxidant enzymes in broiler chicks subjected to embryonic TM. Appendix A includes the mRNA levels of the same antioxidant genes for broiler chicks (TM and controls) not exposed to heat stress during the same timeframe.

*Catalase*. On day 0 (PD 28) of heat stress, TM led to significantly lower catalase mRNA levels in the liver. In the control group, the hepatic mRNA levels of catalase were significantly higher after 5 (PD 33) and 7 (PD 35) days of heat stress compared to day 0, while, in the TM group, the level was significantly higher only after 1 day (PD 29) of heat exposure. The hepatic catalase mRNA level was significantly lower in TM chicks compared to controls after 5 (PD 33) and 7 (PD 35) days of heat stress.

The splenic mRNA level of catalase was not significantly different between TM and control chicks on day 0 (PD 28) of heat stress. However, the level was significantly lower in TM chicks compared to controls after 1 (PD 29) and 7 (PD 35) days of heat stress. Within the control group, the splenic mRNA level of catalase was significantly higher after 1 (PD 29), 5 (PD 33), and 7 (PD 35) days of heat exposure compared to day 0 (PD 28), whereas, in the TM group, the splenic mRNA level of catalase was significantly higher after 3 (PD 31) and 5 (PD 33) days (vs. day 0 (PD 28)).

*NOX4*. In the liver, the mRNA level of *NOX4* was not significantly different between the TM and control groups on day 0 (PD 28) of heat stress. In contrast, the *NOX4* mRNA level was significantly lower in TM chicks compared to controls after 3 (PD 31), 5 (PD 33), and 7 (PD 35) days of heat stress. Within the control group, the mRNA level was significantly increased after 3 (PD 31), 5 (PD 33), and 7 (PD 35) days of heat stress (vs. day 0 (PD 28)), but, in the TM group, the level did not significantly change during heat stress compared to day 0 (PD 28).

Similarly, the splenic mRNA level of *NOX4* was not significantly different between TM and control chicks on day 0 (PD 28) of heat stress. However, the level was significantly lower in the TM group compared to controls after 5 days (PD 33) of heat stress. Within the control group, the mRNA level of *NOX4* was significantly increased after 5 days (PD 33) of heat exposure in comparison with day 0 (PD 28), while in the TM group, the level did not significantly change during heat exposure in comparison with day 0 (PD 28).

*SOD2*. The liver mRNA level of *SOD2* was not significantly different between the TM and control groups on day 0 (PD 28) of heat stress. However, *SOD2* levels were significantly lower in TM chicks compared to controls after 3 (PD 31), 5 (PD 33), and 7 (PD 35) days of heat stress. Within the control group, the hepatic mRNA level of *SOD2* was significantly higher after 7 days (PD 35) of heat stress compared to day 0 (PD 28), while, in the TM group, the level significantly increased only after 1 day (PD 29) of heat exposure (vs. day 0 (PD 28)).

The splenic mRNA level of *SOD2* did not significantly differ between the TM and control groups on day 0 (PD 28) of heat stress. Contrastingly, the splenic level was significantly lower in TM chicks compared to controls after 7 days (PD 35) of heat exposure. Within the control group, the splenic mRNA level of *SOD2* was significantly higher after 7 days (PD 35) of heat stress compared to day 0 (PD 28), whereas, in the TM group, the level was significantly higher only after 5 days (PD 33) of heat exposure (vs. day 0 (PD 35)).

### 3.4. Effect of Post-Hatch Cold Stress on Physiological Parameters of Thermally Manipulated Broilers

Table 4 represents the effects of cold stress for 5 days (PD 32 to 37) on T^C^, BW, and BW gain in thermally manipulated broiler chickens and controls. Application of TM significantly decreased the mortality rate during post-hatch exposure to cold stress (control = 5%; TM = 0). In contrast, cold stress did not significantly affect T^C^, but, in both the TC and TN subgroups, controls exhibited significantly higher T^C^ compared to TM chicks. On day 0 (PD 32) of cold exposure, there was no significant change was observed in BW between the control and TM groups. After 5 days (PD 37) of cold stress, the BW of controls was significantly lower in comparison with control chicks exposed to cold stress. Furthermore, cold stress significantly decreased the BW gain in both the control and TM chicks, although the weight gain was significantly lower in controls compared to TM chicks.

### 3.5. Effect of Post-Hatch Cold stress on mRNA Levels of Antioxidant Enzymes in Thermally Manipulated Broilers

Figure 3 represents the effects of cold stress on the mRNA levels of antioxidant enzymes in the liver, spleen heart and breast muscle of broiler chickens subjected to embryonic thermal manipulation.

*Catalase*. TM did not significantly change the cardiac, hepatic, and muscular mRNA levels of catalase in chicks kept under normal environmental temperatures. However, TM significantly decreased the catalase mRNA level in the spleen. Regarding those chicks exposed to cold stress, the TM group possessed a significantly lower mRNA level of catalase in the liver, spleen, and heart compared to controls.

*NOX4*. In the chicks of the TN subgroup, TM chicks possessed significantly lower splenic, hepatic, and cardiac mRNA levels of *NOX4* compared to controls. Despite this, muscular NOX4 mRNA levels were significantly higher in the TM chicks. Similar results were observed in the chicks exposed to cold stress.

*SOD2*. No significant changes were seen in the hepatic and muscular *SOD2* mRNA levels between the TM and control groups exposed to normal conditions. However, the splenic and cardiac levels of *SOD2* mRNA were significantly higher in controls compared to TM chicks. After cold exposure, the cardiac, hepatic, and splenic mRNA levels of *SOD2* were significantly higher in controls compared to TM chicks, but the level in breast muscle was significantly higher in the TM group.

## 4. Discussion

Oxidative damage is caused by excess reactive oxygen species (ROS), such as superoxide (O_2_^−^) and hydrogen peroxide (H_2_O_2_), which are a necessary product of aerobic metabolism [35]. Heat stress is a major cause of oxidative damage in poultry, and it is associated with a modulation in the expression of antioxidant genes, including catalase, *NOX4*, and *SOD2* [10,13]. Thermal manipulation (TM) has often been suggested as a viable method of improving the acquisition of thermotolerance in heat-stressed broilers [31,36,37,38]. However, the effects of TM and subsequent heat challenge on broiler antioxidant capacity has not been extensively explored. Similarly, a dearth of information exists with regard to the effects of cold stress on broiler gene expression, especially within the context antioxidant gene expression. The objective of the present study was two-fold: it aimed to ascertain the effects of embryonic TM on broilers under conditions of post-hatch heat stress as well as cold stress.

During heat stress, the behavior of broilers is altered as they attempt to decrease their body temperature (T^C^), resulting in myriad negative effects on performance [39]. In the current study, TM was found to result in significantly lower T^C^ on post-hatch days (PD) 1 and 37 and higher body weights (BW) on PD 37 compared to controls. Correspondingly, it has often been reported that TM treatments significantly increased broiler BW [37,40,41] and improved their abilities to regulate their T^C^ in periods of heat challenge [31,32,42]. Lower T^C^ during heat stress also improved feed conversion ratios in TM broilers, and it has been suggested that the lower T^C^ in TM broilers is due to slower metabolic rates as a result of the TM treatment [43].

With regard to post-hatch heat stress, our findings show that TM chickens had significantly decreased mortality rates and BT as well as increased BW compared to controls. Heat stress has been extensively reported to affect broiler physiological parameters [8,44,45]. On a similar note, cold-stressed TM chickens had significantly lower mortality rates than cold-stressed controls. Previously, TM has been found to reduce the mortality rates of broilers during heat challenge [46,47]. Contrastingly, one study reported that heat-stressed TM broilers experienced higher mortality rates than their control counterparts [48]. These differences in findings may be attributed to the fact that there is no one single type of TM treatment, and different studies employ different periods and conditions of TM.

The catalase enzyme is found in the majority of aerobic organisms as well as in some obligate anaerobes [49]. Catalase is responsible for the breakdown of hydrogen peroxide (H_2_O_2_) into oxygen and water, thereby preventing oxidative damage from occurring in a cell [50]. In the present study, catalase expression was significantly modulated in heat- and cold-stressed TM and control chickens. In fact, heat stress resulted in decreased hepatic and splenic catalase expression in TM chickens compared to controls, while cold stress led to significantly lower cardiac, hepatic, and splenic catalase expression levels in the TM group. Compared to controls, heat-stressed TM chickens were previously reported to exhibit decreased catalase mRNA levels [13]. Moreover, broilers were found to exhibit higher levels of catalase activity during acute heat stress, but this antioxidant capacity decreased with age [51]. In female broilers, cardiac catalase activity was reduced after cold stimulation [52].

To maintain homeostasis, the NOX4 enzyme is heavily involved in the oxygen-sensing process, the latter of which causes it to generate significant amounts of ROS [17]. Additionally, *NOX4* over-expression is often associated with oxidative stress in a number of different organs [18,53,54]. In avian muscle cells, ROS production during heat stress and subsequent oxidative damage has been tentatively attributed to *NOX4* up-regulation [55]. In the present study, hepatic and splenic *NOX4* expression levels were significantly lower in heat-stressed TM chickens compared to controls. Similarly, after cold stress, TM chickens exhibited decreased cardiac, hepatic, and splenic but increased muscular *NOX4* expression than that in controls. Hepatic *NOX4* mRNA expression was previously reported to be lower in TM chickens exposed to heat stress compared to controls [13]. In cultured avian cells, heat stress was found to upregulated *NOX4* mRNA expression [55].

The SOD2 enzyme functions to transform the superoxide (O_2_^−^) radical into hydrogen peroxide and water, and it plays an important cytoprotective role against oxidative stress [56]. Our findings indicate that both heat and cold exposure led to generally decreased *SOD2* expression in several organs. Compared to controls, heat-stressed TM chickens displayed lower hepatic and splenic *SOD2* expression levels, while cold-stressed TM chickens showed decreased cardiac, hepatic, and splenic *SOD2* expression. Like *NOX4*, however, muscular *SOD2* expression levels were higher in cold-stressed TM chickens compared to their control counterparts. A previous study found that hepatic *SOD2* expression and enzymatic activity were decreased in TM chickens exposed to heat stress [13]. In contrast, another study found that *SOD* mRNA levels in two broiler strains (Cobb and Hubbard) were unaffected by heat stress [57]. Additionally, *SOD2* levels remained unchanged in avian cell cultures exposed to heat stress [55]. The present findings may suggest that lower levels of oxidative *NOX4* expression may lead to lower expression of the anti-oxidative catalase and *SOD2* genes.

Interestingly, mRNA expression levels of the catalase, *NOX4*, and *SOD2* genes in the breast muscle differed from those in the heart, liver, and spleen in cold-stressed TM chickens. Such inter-organ variation in gene expression is to be expected, as expression varies to a larger degree between organs of a single species than between different species [58]. However, in broilers, the breast muscle in particular has been subject to rapid changes in size and conformation over the past few decades due to the artificial selection pressures applied by the commercial poultry industry [59]. This has resulted in a number of abnormalities and myopathies of the breast muscle that is estimated to affect up to 90% of broilers worldwide [60,61,62]. In fact, broiler breast muscle cells were suggested to constantly undergo hypoxic stress, as the transcriptional profiles of non-stressed broiler breast muscle and heat-stressed layer breast muscle were similar [63].

A number of strengths can be found in the current study. All samples were taken from male Cobb chicks in order to reduce inter-strain and inter-sex genetic variation. Moreover, any non-experimental stress was minimized by ensuring that the rearing and experimental rooms were in close proximity to one another. However, there are some limitations of the present study. Firstly, the effect of TM on the developmental parameters of broiler embryos was not investigated, requiring future research. Secondly, the oxidation levels of lipids, proteins, and DNA in different tissues must still be measured in order to ascertain the final balance of catalase, *NOX4*, and *SOD2* expression. Lastly, the exact impact of TM on embryonic mortality was not considered, which mandates future lines of research in this context.

## 5. Conclusions

Our findings indicate that TM at 39 °C and 65% RH for 18 h/day from days 10 to 18 of embryonic development might result in positive long-lasting effects on broiler antioxidant capacity. Future research should focus on the effects of TM and subsequent thermal challenge on various types of broiler muscle, as the expression dynamics of the breast muscle was found to differ from those of other organs.

## Figures and Tables

**Figure 1 animals-10-00126-f001:**
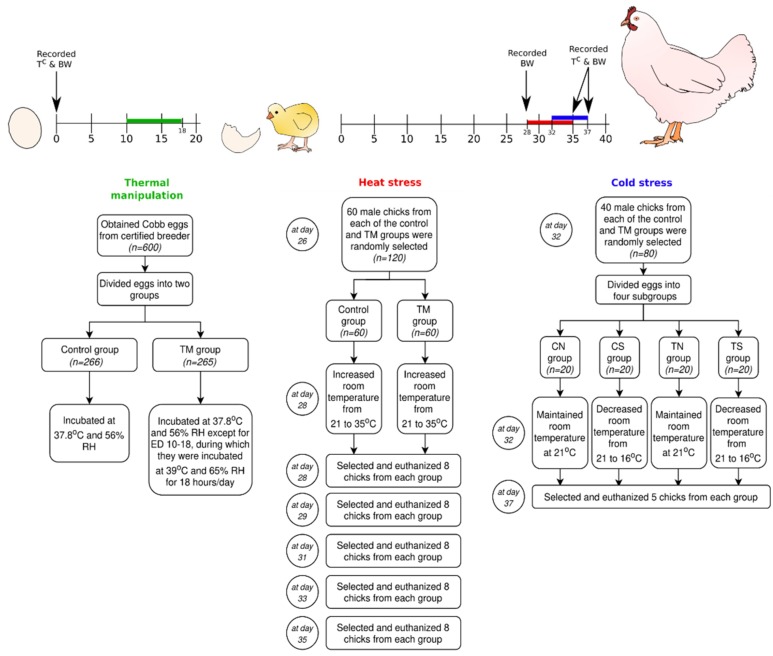
Experimental design showing the three main phases: thermal manipulation, heat stress, and cold stress.

**Figure 2 animals-10-00126-f002:**
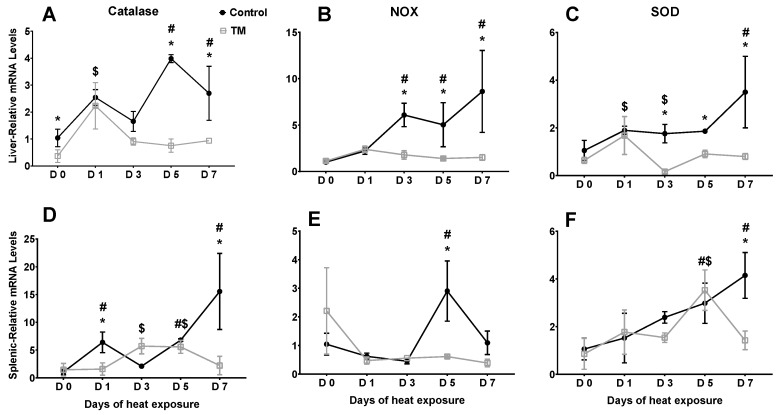
Effect of post-hatch heat stress for 7 days (PD 28 to 35) on the mRNA levels of catalase, *NOX4*, and *SOD2* in the liver (**A**–**C**) and spleen (**D**–**F**) of TM broiler chicks (n = 5). * within the same day, means ± SD of TM and control chicks are significantly different. ^#^ within the control group, means ± SD of non-identical days differ significantly. ^$^ within the TM group, means ± SD of non-identical days differ significantly.

**Figure 3 animals-10-00126-f003:**
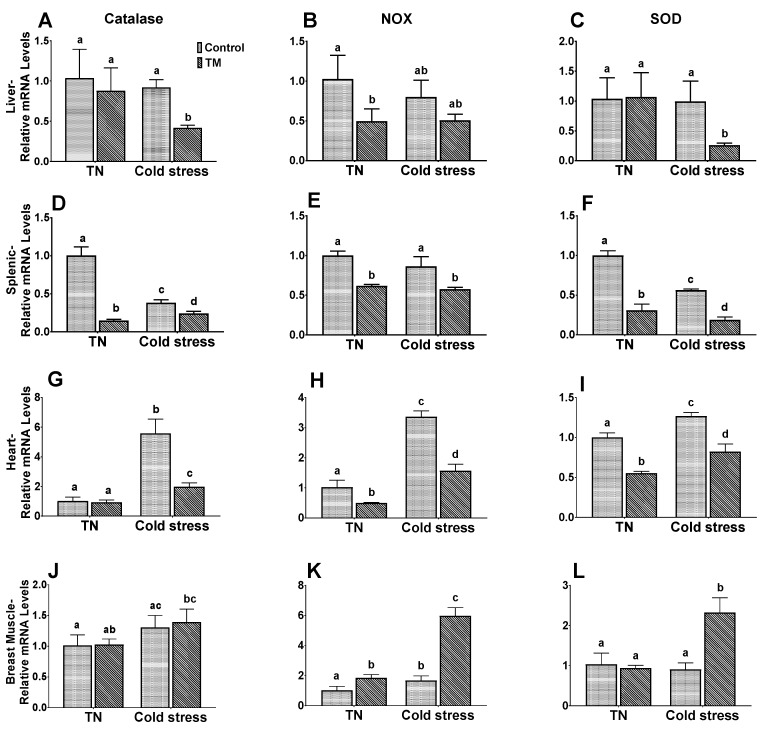
Effect of post-hatch cold stress for 5 days (PD 32 to 37) on the mRNA levels of catalase, *NOX4*, and *SOD2* in the liver (**A**–**C**), spleen (**D**–**F**), heart (**G**–**I**), and breast muscle (**J**–**L**) in TM broiler chickens (n = 5). ^a–d^ means ± SD with non-identical superscripts are significantly different.

**Table 1 animals-10-00126-t001:** Primer sequences used for real-time RT-PCR analysis.

Gene	Forward (5′ to 3′)	Reverse (5′ to 3′)
*NOX4*	CCAGACCAACTTAGAGGAACAC	TCTGGGAAAGGCTCAGTAGTA
*SOD2*	CTGACCTGCCTTACGACTATG	CGCCTCTTTGTATTTCTCCTCT
Catalase	GAAGCAGAGAGGTTCCCATTTA	CATACGCCATCTGTTCTACCTC
28S rRNA	CCTGAATCCCGAGGTTAACTATT	GAGGTGCGGCTTATCATCTATC

**Table 2 animals-10-00126-t002:** Effects of embryonic thermal manipulation (TM) on post-hatch body weight (BW) and cloacal temperature (T^C^) of broiler chickens.

	Post-Hatch Day	Control	TM
T^C^ (°C)	1	39.63 ± 0.24 ^a^	39.48 ± 0.23 ^b^
	37	39.05 ± 0.24 ^a^	38.38 ± 0.22 ^b^
BW (g)	1	44.2 ± 3.8 ^a^	42.7 ± 2.8 ^a^
	37	2302.8 ± 79.7 ^a^	2440 ± 82.5 ^b^

^a,b^ within the same row, means ± SD with non-identical superscripts are significantly different.

**Table 3 animals-10-00126-t003:** Effects of post-hatch heat stress for 7 days (post-hatch day (PD) 28 to 35) on cloacal temperature (T^C^), body weight (BW), and BW gain in broiler chickens subjected to embryonic thermal manipulation and controls.

	Normal Conditions		Heat Stress	
	(21 °C; RH 45%–52%)		(35 °C; RH 45%–52%)	
	Control	TM	Control	TM
T^C^ (°C)	39.65 ± 0.28 ^a^	39.08 ± 0.21 ^b^	41.35 ± 0.24 ^c^	40.15 ± 0.26 ^d^
BW (g)				
Day 0 (PD 28)	1373.3 ± 51.3 ^a^	1675.7 ± 83.8 ^b^	1456.7 ± 82.8 ^a^	1704 ± 74.4 ^b^
Day 7 (PD 35)	1847.9 ± 108.1 ^a^	2108.8 ± 95.8 ^b^	1645 ± 40.6 ^c^	1930 ± 50.2 ^d^
BW gain (g)	474.6 ± 70.9 ^a^	433 ± 39.1 ^a^	188.3 ± 47.3 ^b^	226 ± 34.7 ^b^

^a–d^ within the same row, means ± SD with non-identical superscripts are significantly different.

**Table 4 animals-10-00126-t004:** Effect of post-hatch cold stress for 5 days (post-hatch day (PD) 32 to 37) on cloacal temperature (T^C^), body weight (BW), and body weight gain in broiler chickens subjected to embryonic thermal manipulation and controls.

	Normal Conditions		Cold Stress	
	(21 °C; RH 45%–52%)		(16 °C; RH 45%–52%)	
	Control (CN)	TM (TN)	Control (CS)	TM (TS)
T^C^ (°C)	39.18 ± 0.35 ^a^	38.5 ± 0.34 ^b^	39.3 ± 0.2 ^a^	38.93 ± 0.32 ^ab^
BW (g)				
Day 0 (PD 32)	1717.9 ± 137.5 ^a^	1834 ± 112.8 ^a^	1720 ± 147.5 ^a^	1845.5 ± 119.9 ^a^
Day 5 (PD 37)	2244.3 ± 134.4 ^a^	2281 ± 101.9 ^a^	1993.1 ± 131.3 ^b^	2179 ± 134.8 ^a^
BW gain (g)	526.4 ± 41.1 ^a^	447 ± 42.2 ^b^	273.1 ± 23.9 ^c^	333.5 ± 53.2 ^d^

^a–d^ within the same row, means ± SD with non-identical superscripts are significantly different.

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
