# Peer review of "Embryonic Thermal Manipulation Affects the Antioxidant Response to Post-Hatch Thermal Exposure in Broiler Chickens"

_animals, 2020, doi:10.3390/ani10010126_

Round 1

Reviewer 1 Report

Abstract:

Lines 38 to 39: To read: Catalase, NOX4, and SOD2 expression in the heart, liver, and spleen were decreased in TM chickens----

Line 40: To read: In contrast, the expression of these genes in the breast muscles of the TM group were increased or not affected------

Line 33: Unit for Temperature, °C (insert it in all the text).

Finally, the abstract is deficient in one aspect, that is, you need to capture the results of increase body weight on post-hatch day-37 and decrease cloacal temperature in the TM group.

CS group also, significantly decreased the body weight in both the control and TM groups.

Body weight and cloacal temperature are important production parameters which you should reflect in the abstract.

Introduction:

Line 89: To read: Therefore, the main purpose of the present study was to ---

Materials and Methods:

Line 95: JUST-ACUC: Is there no approval number or code for this study? Because JUST-ACUC is the acronym for Jordan University of Science and Technology's Animal Care and Use Committee.

Line 109: To read: On hatching day, the number of hatched chicks were --

Line 110: To read: one-day-old chicks----

Line 116: To read: whole field experimental period----

Finally, I want to commend your work. It was well carried out, and results presentation is good.

Author Response

Lines 38 to 39: To read: Catalase, NOX4, and SOD2 expression in the heart, liver, and spleen were decreased in TM chickens----

Corrected.

Line 40: To read: In contrast, the expression of these genes in the breast muscles of the TM group were increased or not affected------

Corrected.

Line 33: Unit for Temperature, °C (insert it in all the text).

Inserted.

Finally, the abstract is deficient in one aspect, that is, you need to capture the results of increase body weight on post-hatch day-37 and decrease cloacal temperature in the TM group.

Added.

CS group also, significantly decreased the body weight in both the control and TM groups.

Added.

Body weight and cloacal temperature are important production parameters which you should reflect in the abstract.

Reflected.

Introduction: Line 89: To read: Therefore, the main purpose of the present study was to ---

Corrected.

Materials and Methods: Line 95: JUST-ACUC: Is there no approval number or code for this study? Because JUST-ACUC is the acronym for Jordan University of Science and Technology's Animal Care and Use Committee.

Added the approval number (16/3/3/418).

Line 109: To read: On hatching day, the number of hatched chicks were –

Corrected.

Line 110: To read: one-day-old chicks----

Corrected.

Line 116: To read: whole field experimental period----

Added.

Finally, I want to commend your work. It was well carried out, and results presentation is good.

Thank you for your kind comments.

Reviewer 2 Report

Line 125, define CHS.

Line 143, RNA extracted jejunal mucosa and pancreatic samples was not used in the qPCR analysis?

Line 157, please explain "calculated automatically".

Line 179, section 3.1, for the analysis of day 37, as some of the birds had been exposed to high temperature, and some of them to cold temperature, is it meaningful to test for the cloacal temperate and body weight? The effect is not caused by just TM, but TM + heat / cold stress.

Author Response

Line 125, define CHS.

Removed reference to chronic heat stress (CHS).

Line 143, RNA extracted jejunal mucosa and pancreatic samples was not used in the qPCR analysis?

Removed references to jejunal mucosal and pancreatic samples, the latter of which were reported in another study of ours.

Line 157, please explain "calculated automatically".

Added explanation.

Line 179, section 3.1, for the analysis of day 37, as some of the birds had been exposed to high temperature, and some of them to cold temperature, is it meaningful to test for the cloacal temperate and body weight? The effect is not caused by just TM, but TM + heat / cold stress.

During the rearing period (post-hatch days 1 to 37), the birds were kept under standard conditions, but, for heat and cold exposures, a number of birds were selected randomly and transferred to different rooms. Therefore, the cloacal temperatures and body weights measured on post-hatch day 37 were for birds reared under standard conditions.

Reviewer 3 Report

Animals-663171 - Embryonic thermal manipulation affects the antioxidant response to post-hatch thermal exposure in broiler chickens.

Comments:

Essentially, the manuscript is describing the interference of temperature on incubation of fertile eggs and their viability after hatched.  The manuscript is well written and it was developed in great experimental conditions, as It is possible to verify in the description in the text.

There are only a few points to be cleared before final acceptance: Was mortality recorded during incubation and during rearing phase? Which did incubation phase occurred embryonic mortality and which the percentage? Which was rearing period adopted, If from 1 to 42 days-old or less? If possible, the authors must add another information on development parameters of the chickens hatched in these conditions of thermal manipulation. In my opinion it very important to highlight these data in text.

Author Response

Essentially, the manuscript is describing the interference of temperature on incubation of fertile eggs and their viability after hatched.  The manuscript is well written and it was developed in great experimental conditions, as It is possible to verify in the description in the text.

There are only a few points to be cleared before final acceptance: Was mortality recorded during incubation and during rearing phase?

The mortality percentage was calculated during the post-hatch rearing phase. However, the mortality was not calculated for incubation. Instead, we calculated the hatchability, which is the percentage of fertile eggs that hatch.

Which did incubation phase occurred embryonic mortality and which the percentage?

The exact impact on embryonic mortality was not considered in this study, and this limitation was added to the Discussion section.

Which was rearing period adopted, If from 1 to 42 days-old or less?

The entire rearing period applied in this study was 37 days.

If possible, the authors must add another information on development parameters of the chickens hatched in these conditions of thermal manipulation. In my opinion it very important to highlight these data in text.

In the present study, we only evaluated the hatchability, hatchling weight, and hatchling temperature to investigate the impact of TM on developmental parameters. However, this limitation was added to the Discussion section

Reviewer 4 Report

The manuscript authored by Saleh et al. reported how thermal manipulation during egg incubation affected oxidative and antioxidative gene expression in varied tissues of broiler chickens exposed to heat- or cold-stresses after ~1 month hatching. While the results are potentially interesting to readers in the poultry industry, this reviewer has concerns regarding the study design.

Figure 2B shows that 28 days after hatching, NOX gene expression in the liver was no different between Control and TM before thermal stress; while Figure 3B shows that 37 days after hatching, NOX gene expression in the liver was significantly lower in the TM than in the Control without thermal stress; therefore the conclusion is that age affects hepatic gene expression of NOX differently in TM and Control. This conclusion can be derived using similar logic to other genes in other tissues (i.e., catalase in spleen). Thus to properly evaluate the responses of these genes to heat stress (i.e., to exclude the effects of age), both TM and Control chickens should have data at normal room temperature on D1, D3, D5,and D7 in the 7-day experiment, meaning Figure 2 is missing two experimental arms (i.e., Control-TN and TM-TN). This is a fetal flaw of this study.

Furthermore, lower levels of oxidative NOX expression may lead to lower expression of anti-oxidative catalase and SOD, and to measure the final balance, it would be desirable to measure oxidation levels of lipid, protein, or DNA in varied tissues.

Author Response

The manuscript authored by Saleh et al. reported how thermal manipulation during egg incubation affected oxidative and antioxidative gene expression in varied tissues of broiler chickens exposed to heat- or cold-stresses after ~1 month hatching. While the results are potentially interesting to readers in the poultry industry, this reviewer has concerns regarding the study design.

Figure 2B shows that 28 days after hatching, NOX gene expression in the liver was no different between Control and TM before thermal stress; while Figure 3B shows that 37 days after hatching, NOX gene expression in the liver was significantly lower in the TM than in the Control without thermal stress; therefore the conclusion is that age affects hepatic gene expression of NOX differently in TM and Control. This conclusion can be derived using similar logic to other genes in other tissues (i.e., catalase in spleen). Thus to properly evaluate the responses of these genes to heat stress (i.e., to exclude the effects of age), both TM and Control chickens should have data at normal room temperature on D1, D3, D5,and D7 in the 7-day experiment, meaning Figure 2 is missing two experimental arms (i.e., Control-TN and TM-TN). This is a fetal flaw of this study.

We carried out the aforementioned experimental arms but found no differences in gene expression between them. To avoid the risk of over-complicating the results, we set day 1 of heat stress (post-hatch day 28) as the reference to which the other days were compared.

Furthermore, lower levels of oxidative NOX expression may lead to lower expression of anti-oxidative catalase and SOD, and to measure the final balance, it would be desirable to measure oxidation levels of lipid, protein, or DNA in varied tissues.

This limitation was added to the Discussion section.

Reviewer 5 Report

The manuscript “Embryonic thermal manipulation affects the antioxidant response to post-hatch thermal exposure in broiler chickens” deals with a noteworthy topic presenting novel information on this actual topic. The results enlighten possibilities to improve broiler management in hot climate conditions, also presenting new information on the cold stress of broilers. The described paper is, therefore, relevant to the readers of the journal.

The manuscript is fluently written and easy to read. However, materials and methods require more detailed information. Also, the discussion appears quite superficial and thus deeper and more comprehensive discussion would be appreciated. Furthermore, in the discussion, a paragraph focusing on the strengths and limitations would improve the quality of the paper. Implications for future research should also be clearly discussed, instead of only briefly mentioning them within conclusions.

I would expect the authors’ clarification regarding several points before I can recommend this paper for publication. More detailed comments can be found in the attachment.

Author Response

The manuscript “Embryonic thermal manipulation affects the antioxidant response to post-hatch thermal exposure in broiler chickens” deals with a noteworthy topic presenting novel information on this actual topic. The results enlighten possibilities to improve broiler management in hot climate conditions, also presenting new information on the cold stress of broilers. The described paper is, therefore, relevant to the readers of the journal.

The manuscript is fluently written and easy to read. However, materials and methods require more detailed information. Also, the discussion appears quite superficial and thus deeper and more comprehensive discussion would be appreciated. Furthermore, in the discussion, a paragraph focusing on the strengths and limitations would improve the quality of the paper. Implications for future research should also be clearly discussed, instead of only briefly mentioning them within conclusions.

I would expect the authors’ clarification regarding several points before I can recommend this paper for publication.

Line 30: The abbreviation TM requires explanation in the abstract, even though the explanation can be found in the simple summary.

Added explanation in abstract.

Lines 99-100: The authors mention that damaged eggs were excluded but do not inform how many eggs actually were included in the study. The reader should calculate that form the figure, and that is annoying. Please provide information about the number of eggs remaining in the study.

Added number of eggs to the text.

Line 112: Only the mortality rate was recorded but not the reasons. Why?

Dead chickens were histopathologically examined, but no significant obvious findings were reported. This information was added to the Methods section.

Line 114: What was the relative humidity? It seems a bit confusing that, at some points, relative humidity is mentioned (for example line 126) but not always. Both temperature and relative humidity are important housing conditions from broiler’s welfare point of view. The authors should inform also relative humidity along temperature information.

Added information about relative humidity.

Figure 1: This is a nice way to show study design.

Thank you for your kind comment.

Line 115: The sentence “Starting from post-hatch day 24 until 37, the temperature was maintained at 21oC.” probably refers to the control group? The text should be clarified.

Clarified text.

Lines 124 and 134: Were the selected birds kept in groups of 60 and 40? Earlier (line 113), the authors have explained that the chicks were kept in groups of 10. The text requires clarification.

Added clarification.

Line 143: The authors are referring to jejunal and pancreatic samples, but these organs were not mentioned earlier, on lines 130 and 139-140, when sample collection was explained. The description of sample collection should be clarified.

Removed references to jejunal mucosal and pancreatic samples, the latter of which were reported in another study of ours.

Why the age and duration for heat and cold stress were chosen to be different? The authors should provide some reasoning behind these choices. Answer: The reason is that the research team was not able to carry out the two experiments in the same period with high accuracy. That’s why we preferred to perform each experiment independently. As the same rearing conditions are similar at time frame of both heat and cold experiments (21 oC and 45-55% RH). It was no matter which experiments we performed first. The other thing our interests was examined the possible effects of thermal manipulation on the improved thermotolerance acquisition to heat stress (heat and cold stress) in chicken on the market age (32-42 days post-hatching) as they are more susceptible to acute or chronic heat stress.

Line 167: As far as I can see, hatchability is mentioned for the first time in here. I would expect to see some details about how the hatchability was measured already earlier in the materials and methods section.

Added information about how hatchability was measured.

- Line 181: Instead of “No ... both ... and ...”, the text might be more fluent if written like: “No significant effect was observed in either ... or ...”

Corrected.

Line 190: Surely, broiler is not an adult bird at slaughter age. Please use another word instead of “adult”.

Removed the term “adult”.

Lines 191-192: Table 3 presents the results of thermally manipulated broilers and controls, does it not? The text should be rephrased.

Rephrased.

Line 196: It would be easier for the reader, it the authors would inform both the day of heat stress and the age of the birds in days; for example “on day 7 of heat stress (at 35 days of age)”.

Included both heat stress and age of birds in days.

In Table 3, the first column Control, rows Day 0 and Day 7, the results should be in line with the results on rows above and below.

Corrected.

The table should include information about what conditions (temperature and relative humidity) were used in normal situation and in heat stress situation.

Included.

Line 253: It is not recommendable to start a sentence with an abbreviation.

Corrected.

Table 4 presents the results of thermally manipulated broilers and controls, does it not? The text should be rephrased, both in the table legend and within the text (lines 252-253).

Corrected.

Lines 296-297: The sentence requires a reference.

Added references.

Lines 304-308: The authors should deepen the discussion by considering the reasons for their results instead of only referring to earlier similar results. This comment applies to many other paragraphs in the Discussion section. For example, on lines 319-325, the authors could explain the role of catalase, NOX4, SOD2 generally and during the heat or cold stress. Why are these enzymes important? I am puzzled because, in the abstract and introduction, the authors are talking a lot about oxidative stress, but not once are they referring to oxidative stress in the discussion. The discussion requires improvement; it needs to be deeper and more comprehensive.

Discussion was improved and made more comprehensive according to the reviewer’s guidelines. Moreover, a paragraph was added at the end of the Discussion regarding the strengths and limitations of this study.

- Line 312: Delete the extra blank space before the dot.

Deleted.

- Line 357: Please explain are the long-lasting effects positive or negative.

Explained that the effects are positive.

Round 2

Reviewer 4 Report

In the authors' response, it was mentioned that the "aforementioned" experimental arms were conducted. It would be convincing if these data is provided as supplemental figures or tables. 

Author Response

n the authors' response, it was mentioned that the "aforementioned" experimental arms were conducted. It would be convincing if these data is provided as supplemental figures or tables. 

Response: supplemental tables are added.

Round 3

Reviewer 4 Report

The manuscript is acceptable for publication.

Author Response

Thank you very much for your kind help. the manuscript has been reviewed for spell checks.